Corrected: Publisher correction

# Aerodynamic generation of electric fields in turbulence laden with charged inertial particles

M. Di Renzo [1,2] & J. Urzay[1]

Self-induced electricity, including lightning, is often observed in dusty atmospheres. However, the physical mechanisms leading to this phenomenon remain elusive as they are remarkably challenging to determine due to the high complexity of the multi-phase turbulent flows involved. Using a fast multi-pole method in direct numerical simulations of homogeneous turbulence laden with hundreds of millions of inertial particles, here we show that mesoscopic electric fields can be aerodynamically created in bi-disperse suspensions of oppositely charged particles. The generation mechanism is self-regulating and relies on turbulence preferentially concentrating particles of one sign in clouds while dispersing the others more uniformly. The resulting electric field varies over much larger length scales than both the mean inter-particle spacing and the size of the smallest eddies. Scaling analyses suggest that low ambient pressures, such as those prevailing in the atmosphere of Mars, increase the dynamical relevance of this aerodynamic mechanism for electrical breakdown.

[1] Center for Turbulence Research, Stanford University, Stanford, CA 94305, USA. [2] Dipartimento di Meccanica, Matematica e Management, Politecnico di Bari, 70125 Bari, Italy. Correspondence and requests for materials should be addressed to J.U. (email: jurzay@stanford.edu)

The transport of charged particles in turbulent flows is a problem of interest for the study of conditions leading to atmospheric electricity phenomena in terrestrial and extra-terrestrial environments. For instance, it is known that desert sandstorms on Earth routinely bear relatively large electric fields of order 5–150 kV m$^{-1}$ [1–6]. This phenomenon involves sand or dust particles, which, upon being lifted off by the wind, become triboelectrically charged by collisions near the ground within the saltation layer, where the particle density is large, with small and large particles tending to be negatively and positively charged, respectively, as a result of differential transfer of free electrons from collisions between particles of different sizes [5,7–10].

A paradoxical example of these interactions is the Martian atmosphere, where electricity phenomena is expected to be important [11]. The persistent layer of dust that covers the surface expanse of Mars can be easily lifted and dispersed by local weather phenomena such as dust devils and regional storms [12]. The dust particles accumulate in clouds that can morph into global storms known to encircle the entire planet for as long two Earth years [13]. To compound this extreme weather, the prevailing low pressures of order 10 mbar in the $CO_2$-rich Martian atmosphere may favor electric discharges because of the resulting much lower values of the breakdown electric field, which is of order 5–25 kV m$^{-1}$, as opposed to the 3 MV m$^{-1}$ observed on Earth at sea level [14–16]. As a result, the dust clouds are thought to be an excellent brewing environment for electric discharges that might pose risks to Mars surface exploration instruments and crews [12,15–17]. In contrast to other planets in the Solar System, however, conclusive measurements of atmospheric electricity phenomena in Mars are lacking [18], more so after the recent crash of ESA's Schiaparelli Mars lander in 2016 that carried onboard a dust analyzer to elucidate some of these unknowns. Although particle-to-ground collisions are generally known to lead to tri-boelectric charging very close to the ground [5,7], the occurrence of mesoscopic electric discharges in the gas above must however rely on aerodynamic mechanisms to spatially segregate the charged airborne dust. One of such mechanisms is formulated in the present work.

In this investigation, direct numerical simulations (DNS) are employed to quantify the electric fields generated due to the aerodynamic segregation of a dilute suspension of bi-disperse, oppositely charged, inertial point particles laden in statistically stationary, homogeneous-isotropic turbulence. The particular dispersing agent studied here is related to the vortical inter-mittency inherent to turbulent flows, whereby the particles are centrifuged away from vortices and accumulate in interstitial strain-rate-dominated regions, as schematically depicted in Fig. 1, in a phenomenon that is usually referred to as preferential concentration [19–22]. The analysis assumes that the distribution of airborne particles is electrically pre-charged as a result of the frequent collisions against the ground that prevail within the saltation layer [5,7], which is located far below the present simulation domain and extends up to centimeters on Earth [23] and meters on Mars [24]. Despite the corresponding exponential decrease in particle concentration away from the ground [23], which facilitates the analysis by relegating mid-air collisions to a second-order effect [25] (see also the Methods section), the necessary consideration of long-range electric forces between the hundreds of millions of particles that need to be tracked in the flow field makes the numerical integration of the problem particularly challenging, as it involves the resolution of an $N$–body problem that is additionally coupled with the multi-scale dynamics of turbulence, with $N = O(10^8)$, the number of particles considered here. To circumvent these difficulties, the present work couples the fast multi-pole method (FMM) [26] with the turbulent flow calculation, thereby reducing the simulation cost from $O(N^2)$ to $O(N \log N)$.

An aerodynamic mechanism for the production of long-wavelength electric fields in the carrier gas is formulated here that is unrelated to gravity and collisions, and is based on the segregation of particles by turbulence depending on their inertia. The electric-charge sign is imposed to be negative for small particles and positive for large particles in accord with experimental observations, including characterizations of Martian dust simulants [9,12,14,17]. The characteristic length associated with the resulting electric fields is much larger than the mean inter-particle distance and the smallest size of the turbulent eddies. Substitution of characteristic dust storm parameters suggest that the mechanism may be capable of producing electric fields of order 40 kV m$^{-1}$ in rarefied atmospheres. These results contribute to the general understanding of electric effects in multi-phase turbulent flows of relevance for space exploration, including the prediction of electricity phenomena in dusty planetary atmospheres.

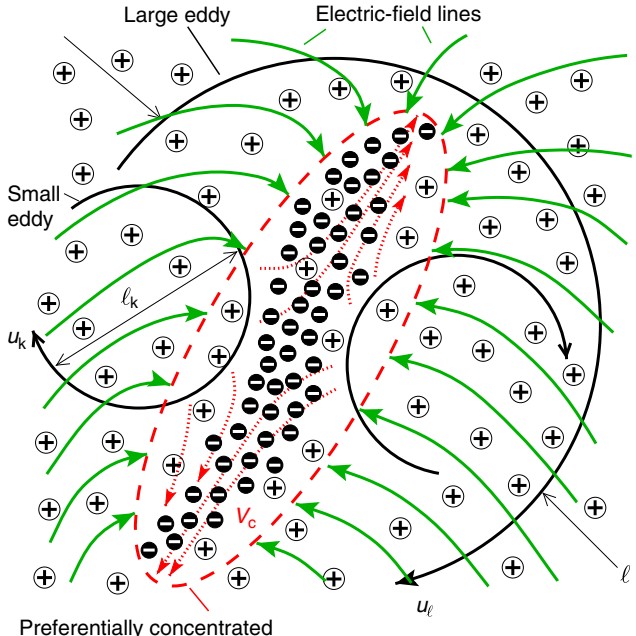

**Fig. 1** Aerodynamic generation of electric fields by turbulence. The turbulence preferentially concentrates negatively charged small particles in interstitial regions between vortices, where the strain rate is large (schematics, not to scale). Conversely, the larger, positively charged particles are comparatively more ballistic, and as a result they are more uniformly distributed than the smaller, negatively charged particles. The large imbalance of charge created within the cloud leads to incoming long-range electric-field lines

## Results

**Characteristic dimensionless parameters**. A qualitative description of the dispersed-phase formulation and the associated dimensionless parameters is given in this section. Quantitative descriptions, along with more detailed explanations of the formulation for both phases, are discussed in the Methods section.

In order to understand the physical processes involved in the generation of electric fields in the present work, it is convenient to outline first some particular limits employed in the simulations. For instance, the dust particles considered here are small compared with the smallest turbulent eddies in such a way that $a_p/\ell_k \ll 1$, where $\ell_k$ is the Kolmogorov length and $a_p$ is the

radius of the particle. Two classes of particles are considered that have different diameters depending on their charge sign, with the positively charged particles being larger than the negatively charged ones, $a_+ > a_-$. In turbulent flows, the Kolmogorov length differs from the size of the large eddies $\ell$ by a factor inversely proportional to the Reynolds number $\mathrm{Re}_\ell = u_\ell \ell / \nu \gg 1$, namely $\ell_k / \ell \sim \mathrm{Re}_\ell^{-3/4} \ll 1$, with $u_\ell$ the integral fluctuation velocity and $\nu$ the kinematic viscosity. Additionally, the density of the dust particles $\rho_p$, which are typically made up of silica[12], is assumed to be much larger than that of the carrier gas $\rho$, namely $\rho_p / \rho \gg 1$. The effects of the particles on the carrier phase are negligible since the mass-loading ratios employed in the simulations for both particle classes, $\alpha = (4/3)\pi \rho_p n_0 a_p^3 / \rho$, are much smaller than unity, with $n_0$ the mean number density of particles, which is the same for both classes. The characteristic Reynolds number of the relative flow motion around the particles is assumed to be small, in such a way that the viscous force acting on the particles is the Stokes drag. Correspondingly, the second Newton's law for each particle can be written

$$\frac{4}{3}\pi \rho_p a_p^3 \frac{d\mathbf{u}_p}{dt} = 6\pi \mu a_p (\mathbf{u} - \mathbf{u}_p) + \mathbf{F}_p, \quad p = 1, \ldots, N, \quad (1)$$

for every particle, with bold symbols denoting vectors. In Eq. (1), $\mu$ is the dynamic viscosity, and

$$\mathbf{F}_p = q_p \mathbf{E}_p \quad (2)$$

is the electric force on the p-th particle, where $q_p$ is the particle charge and $\mathbf{E}_p$ electric field internally generated at $\mathbf{x}_p$ by the surrounding particles. In particular, two values of electric charge, denoted as $q_-$ and $q_+$, are utilized for the small and large particles, respectively, which are equal in magnitude but opposite in sign ($q_+ = -q_-$), in such a way that the particles form an electroneutral system in the mean, $n_0 q_+ + n_0 q_- = 0$, and the net flux of the electric field through the boundaries is zero. Additionally, $\mathbf{u}_p$ and $\mathbf{u}$ are the particle velocity and local fluid velocity, respectively. Specifically, $\mathbf{u}_p$ is related to the particle position $\mathbf{x}_p$ through the trajectory equation

$$d\mathbf{x}_p / dt = \mathbf{u}_p, \quad p = 1, \ldots, N. \quad (3)$$

Upon normalizing $d\mathbf{u}_p/dt$, $\mathbf{u} - \mathbf{u}_p$ and $\mathbf{F}_p$, the non-dimensional version of Eq. (1) renders useful information as follows. Consider nondimensionalizing the slip velocity $\mathbf{u} - \mathbf{u}_p$ with the fluctuation velocity of the Kolmogorov eddies $u_k$, and the acceleration of the particles $d\mathbf{u}_p/dt$ with the acceleration of the Kolmogorov eddies $u_k/t_k$, where $t_k = \ell_k / u_k = \ell_k^2 / \nu$ is the corresponding turnover time. To complete the normalization, a characteristic scale of the electric force is obtained from the Gauss law

$$\nabla \cdot \overline{\mathbf{E}} = (n_+ - n_-) q_+ / \epsilon_0, \quad (4)$$

where $n_+$ and $n_-$ denote, respectively, the local number densities of positively and negatively charged particles, $\epsilon_0$ is the vacuum permittivity, and $\overline{\mathbf{E}}$ is a homogenized electric field that in the present work is only referred to for illustration, is never employed to compute the electric force in Eq. (1), and is subject in the notation to an overbar symbol for reasons that will become clearer later in the text. The exact form of the electric force employed in the simulations, which does not involve homogenization and makes use of the FMM approach in ref. [26] to handle the $N$–body problem, is discussed in the Methods section. In idealized conditions where $n_0$ were sufficiently large for a hypothetical continuum limit to hold in the dispersed phase, Eq. (4) would suggest the scaling $\mathbf{E}_p \sim \overline{\mathbf{E}} \sim n_0 \ell_k q_+ / \epsilon_0$ for the electric field when it is assumed that the characteristic charge

variations are of order $n_0 q_+$, and that the characteristic length for the variations of the electric field is of order $\ell_k$ (both of which are underestimates as evidenced by the numerical results presented below but prove to be convenient for the scaling purposes of this section). Correspondingly, the characteristic scale of the electric force is $\mathbf{F}_p = q_p \overline{\mathbf{E}} \sim n_0 \ell_k q_+^2 / \epsilon_0$. Using these scales, the non-dimensional form of Eq. (1) becomes

$$\frac{d\mathbf{u}_p}{dt} = \frac{1}{\mathrm{St}_{k,\pm}^{(ae)}}\left[\mathbf{u} - \mathbf{u}_p \pm \mathrm{St}_{k,\pm}^{(el)} \mathbf{F}_p\right], \quad (5)$$

where the $\pm$ sign becomes $+$ for positively charged particles and $-$ for negatively charged ones. In addition,

$$\mathrm{St}_{k,\pm}^{(ae)} = t_{a,\pm} / t_k \quad (6)$$

is the aerodynamic Stokes number, with $t_{a,\pm} = (2/9)(\rho_p/\rho)a_\pm^2/\nu$ the characteristic acceleration time of the particles, and

$$\mathrm{St}_{k,\pm}^{(el)} = u_{el,\pm} / u_k \quad (7)$$

is the electric Stokes number, with $u_{el,\pm} = n_0 \ell_k q_+^2 / (6\pi \mu \epsilon_0 a_\pm)$ the electromigration velocity. The parameters Eq. (6) and Eq. (7) are central to the structure of the ensuing electric field, as discussed below.

**The mechanism of turbulence-driven electric fields.** A sketch is provided in Fig. 1 that illustrates the local charge imbalance produced by turbulence and the subsequent generation long-wavelength electric fields. This occurs, for instance, when the negatively charged small particles preferentially concentrate while positively charged large particles do not preferentially concentrate or do so much less intensely (note however that the opposite situation, namely a preferentially concentrated cloud of positively charged particles surrounded by a uniform suspension of negatively charged particles, would lead to the same phenomenon). In the absence of electric fields (i.e., $\mathrm{St}_{k,\pm}^{(el)} = 0$), the disparity between preferential concentration levels of both classes of particles is attained when the conditions

$$\mathrm{St}_{k,-}^{(ae)} \sim 1 \quad \text{and} \quad \mathrm{St}_{k,+}^{(ae)} \gg 1 \quad (8)$$

are satisfied[20,22]. Specifically, the first condition in Eq. (8) states that both the acceleration and slip velocity of the negatively charged small particles are of order unity in Kolmogorov units $u_k/t_k$ and $u_k$, respectively. As a result, these particles move with the large eddies of size $\ell$ but slip with velocities of order $u_{slip,-} \sim u_k$ on the small ones of size $\ell_k$, which bear the strongest levels of vorticity in the flow. Because of their slippage, negatively charged particles preferentially concentrate in the interstitial high-strain regions between those small eddies, as sketched in Fig. 1. In contrast, the second condition in Eq. (8) indicates that the characteristic slip velocity of the positively charged particles is of order $u_{slip,+} \sim (\epsilon t_{a,+})^{1/2} = u_k [\mathrm{St}_{k,+}^{(ae)}]^{1/2} \gg u_k$, where $\epsilon = u_\ell^2/t_\ell = u_k^2/t_k$ is the turbulent dissipation and $t_\ell = \ell/u_\ell$ is the integral time of the turbulence[27]. Correspondingly, the positive particles are ballistic to eddies of sizes ranging from $\ell_k$ to $\ell_k[\mathrm{St}_{k,+}^{(ae)}]^{3/2} \gg \ell_k$, thereby becoming more uniformly distributed in space than the negative ones, as sketched in Fig. 1. The result is an aerodynamic mechanism that segregates negative charges into clouds in an environment of approximately uniformly distributed positive charges.

The classic portrayal representation of preferential concentration in particle-laden turbulence outlined above is not fundamentally altered if the electromigration velocity $u_{el,\pm}$ is smaller than the characteristic slip velocity $u_{slip,\pm}$. In the regime of

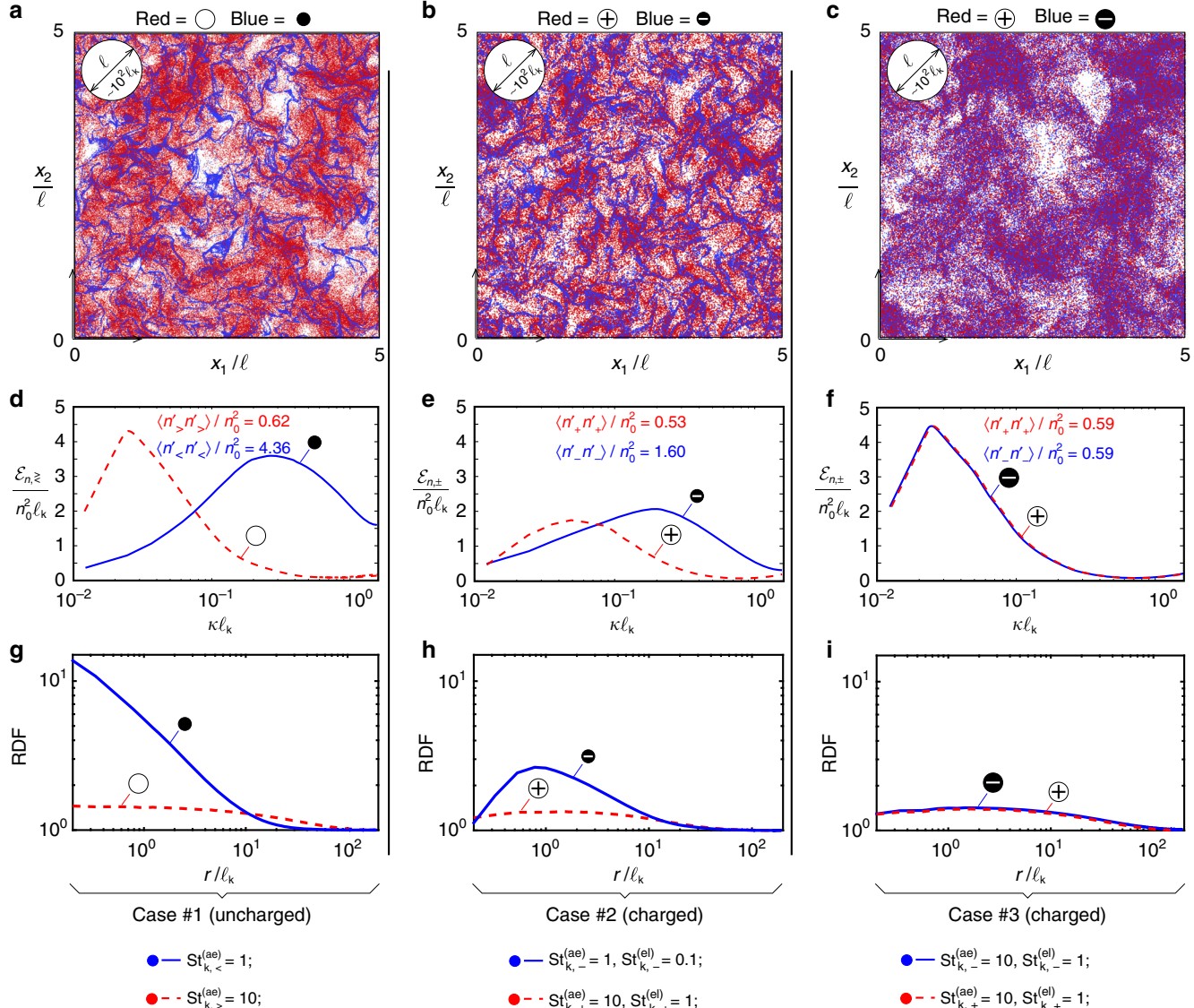

**Fig. 2** Electric effects on the spatial distribution of particles. **a–c** Instantaneous spatial distributions of particles contained in a constant-$x_3$ slice of thickness equal to the Kolmogorov length $\ell_k$. **d–f** Ensemble-averaged spectral energy $\mathcal{E}_n$ of the concentration fluctuations as a function of the wavenumber $\kappa$. **g–i** Ensemble-averaged radial distribution functions (RDFs) as a function of the radial separation distance $r$. The figure includes the uncharged case #1 (**a**, **d**, **g**), the charged case #2 (with negatively charged small particles being preferentially concentrated; **b**, **e**, **h**), and the charged case #3 (with none of the two classes being preferentially concentrated; **c**, **f**, **i**). The integral length $\ell$ and its equivalent size in Kolmogorov units (~100$\ell_k$) are provided for convenience in the left upper corner of **a–c**

interest, the present simulations satisfy conditions leading to such weak electric interactions, in that the electric Stokes numbers are

$$St_{k,-}^{(el)} < 1 \quad \text{and} \quad St_{k,+}^{(el)} < [St_{k,+}^{(ae)}]^{1/2}. \qquad (9)$$

Specifically, both conditions in Eq. (9) ensure that the electric charge carried by each particle is not sufficiently large to induce electric fields capable of causing frequent agglomeration or large deviations from the trajectories induced by the interplay between inertia and Stokes drag in Eq. (1). It is shown below that the overall effect of the electric field on preferential concentration is to decrease it for negatively charged particles and increase it for the positively charged large particles, in such a way that the mechanism of generation of mesoscopic electric fields depicted in Fig. 1 is self-regulating.

Because of the ellipticity associated with the Gauss law (Eq. (4)) in electrostatic conditions, in which $\overline{\mathbf{E}}$ is irrotational, the ensuing electric field that enters the cloud of negatively charged particles necessarily varies along distances longer than the characteristic size $L_c \sim V_c^{1/3}$ of the cloud, where $V_c = O(\ell_k^3)$ is a control volume surrounding it. This effect can be qualitatively understood by a volumetric integration of Eq. (4) within a sufficiently large control volume $V_{el} \gg V_c$ such that the resulting flux of electric field becomes negligible due to the electroneutralization of charge inside. Since the positively charge particles are rather uniformly distributed in the vicinity of the cloud in comparison with the negatively charged ones, the volume integral of the first term on the right-hand side of Eq. (4) can be approximated as $n_0 q_+ V_{el}/\epsilon_0$. In contrast, the negatively charged small particles are mostly concentrated within the cloud in $V_c$, and as a result the

volumetric integral of the second term on the right-hand side of Eq. (4) yields $n_c q_+ V_c/\epsilon_0$, where $n_c \sim C n_0$ is a characteristic number-density fluctuation in the cloud, which, as a result of particle accumulation, is much larger than the mean number density $n_0$ typically by a factor $C$ of order 100. Equating both of these estimates to render zero flux of the electric field, the relation $L_{el}/L_c = C^{1/3} \sim 5$ is obtained, where $L_{el} \sim V_{el}^{1/3}$ is the characteristic length associated with the variations of the mesoscopic electric field. Spectral analyses of the electric field are provided below that ratify these considerations.

**Self-regulating dynamics.** Three different simulation cases are analyzed below that correspond to uncharged (case #1) and charged (case #2) conditions with preferential concentration of small particles, as well as charged conditions under no significant preferential concentration of any of the two classes of particles (case #3). The reader is referred to the Methods section for further descriptions of each case.

The electric field generated by the collective effect of the charged particles is self-regulating, in that it tends to decrease the baseline preferential concentration levels in uncharged calculations at the same aerodynamic Stokes numbers. This is visualized by comparing the instantaneous spatial distributions of particles shown in Fig. 2a, b. The uncharged case #1 in Fig. 2a is characterized by sharp filamentous structures of the preferentially concentrated small particles, which are surrounded by more uniformly distributed particles belonging the other class. In contrast, the charged case #2 in Fig. 2b leads to thicker cloud patterns for the preferentially concentrated, negatively charged small particles, and to a decrease in spatial uniformity for the positively charged large particles. Similar electric effects on preferential concentration have been suggested in early work[28–30] albeit for mono-disperse suspensions of much fewer particles in flows at much lower Reynolds numbers.

The mitigation effect mentioned above is quantified by a spectral analysis of the particle number-density fields, as shown in Fig. 2d, e. The calculations are based on the spectrum $\mathcal{E}_n$ of the energy of the concentration fluctuations, obtained by spherically averaging the multiplication of the fast-Fourier transform of the number density $n$ for each class, in such a way that the integral of the spectrum along the wavenumber axis is equal to the variance of the number density $\langle n' n' \rangle$, where the angular brackets indicate volume averaging over the entire computational domain. In particular, the peak of the spectrum of the small particles in the uncharged case #1 in Fig. 2d is displaced toward high wavenumbers and leads to a larger variance of the number-density field in comparison with the corresponding quantities for the large particles (see legend in Fig. 2d, e). In contrast, in the charged case in Fig. 2e, the variance of the number density of negatively charged small particles is comparatively decreased, while the spectrum peak is displaced toward larger scales, thereby indicating a decrease in preferential concentration due to electric effects. Note however that the opposite trend in the spectrum peak is observed for the positively charged large particles, which is displaced toward smaller scales, indicating the occurrence of finer-grained patterns in the concentration field of this class relative to case #1, although this effect is counteracted by a smaller variance as a result of Coulombic repulsion.

Analogous conclusions are provided by the radial distribution functions (RDFs) shown in Fig. 2g, h, which are defined as the number density of particles in the volume of a finite-thickness spherical shell located at radial distance $r$ from the test particle, divided by the total number density of particle pairs in the spherical volume $4\pi r^3/3$[31]. The large values of the RDF for the

small particles at short distances in the uncharged case #1 in Fig. 2g indicate a high probability of encountering other particles of the same class in the vicinity due to preferential concentration. In contrast, that portion of the RDF decreases significantly in the charged case #2 in Fig. 2h. Specifically, the non-monotonicity of both RDFs in the charged case #2 in Fig. 2h is a consequence of the Coulombic repulsion of particles with the same charge sign that suppresses the occurrence of short separation distances[32]. The discussion of the charged but much more dispersed suspension in Fig. 2c–i leads to similar conclusions as those outlined above, including the non-monotonicity aspect of the RDFs of both particle classes.

The electric effects on preferential concentration described above can be rationalized by taking the divergence of the Eulerian version of Eq. (5), obtained by replacing the time derivative $d/dt$ by the material derivative $D/Dt = \partial/\partial t + \mathbf{u}_p \cdot \nabla$, thereby yielding the expression

$$\frac{D}{Dt}\left(\nabla \cdot \mathbf{u}_p\right) - 2Q_p = \frac{1}{St_{k,\pm}^{(ae)}}\left[\nabla \cdot \mathbf{u}_p \pm St_{k,\pm}^{(el)} \nabla \cdot \mathbf{F}_p\right], \quad (10)$$

where $Q_p = (1/4)\left(\boldsymbol{\omega}_p \cdot \boldsymbol{\omega}_p - 2\mathbf{S}_p : \mathbf{S}_p\right)$ is the second invariant of the particle velocity-gradient tensor. In this formulation, $\mathbf{S}_p = (1/2)\left(\nabla \mathbf{u}_p + \nabla \mathbf{u}_p^T\right)$ is the strain rate of the particle velocity field, and $\boldsymbol{\omega}_p = \nabla \times \mathbf{u}_p$ is the associated vorticity. Of particular interest are flow conditions where $\nabla \cdot \mathbf{u}_p < 0$, which correspond to accumulation of particles along pathlines as dictated by the mass conservation equation for each class,

$$\frac{1}{n_\pm}\frac{Dn_\pm}{Dt} = -\nabla \cdot \mathbf{u}_p > 0, \quad (11)$$

where $n_\pm$ has been normalized with $n_0$. At small Stokes numbers, the acceleration of the particles resembles the fluid acceleration, and as a result the particle velocity becomes

$$\mathbf{u}_p \simeq \mathbf{u} - St_{k,\pm}^{(ae)}\frac{D\mathbf{u}}{Dt} \pm St_{k,\pm}^{(el)}\mathbf{F}_p, \quad (12)$$

with $D/Dt \simeq \partial/\partial t + \mathbf{u} \cdot \nabla$ to leading order. Upon substituting Eq. (12) into Eq. (10), the simplified expression

$$\nabla \cdot \mathbf{u}_p \simeq 2St_{k,\pm}^{(ae)}Q \pm St_{k,\pm}^{(el)}\left(n_+ - n_-\right) \quad (13)$$

is obtained, where use of Eq. (2) and of the homogenization approximation $\mathbf{E}_p \sim \overline{\mathbf{E}}$ has been made, with $\nabla \cdot \overline{\mathbf{E}} \sim n_+ - n_-$ as prescribed by the dimensionless version of the Gauss law (Eq. (4)). In Eq. (13) $Q = (1/4)(\boldsymbol{\omega} \cdot \boldsymbol{\omega} - 2\mathbf{S} : \mathbf{S})$ represents the second invariant of the carrier-phase velocity-gradient tensor defined in terms of the strain-rate tensor $\mathbf{S} = (1/2)(\nabla \mathbf{u} + \nabla \mathbf{u}^T)$ and the vorticity $\omega = \nabla \times \mathbf{u}$.

In view of Eq. (13), the relative importance of aerodynamic and electric effects in shaping the particle concentration fields is quantified by the Stokes-number ratio $St_{k,\pm}^{(ae)}/St_{k,\pm}^{(el)}$. For neutral particles, $St_{k,\pm}^{(el)} = 0$, and Eq. (13) leads to the classic conclusion that flow regions where straining prevails over vortical motion ($Q < 0$) tend to be preferentially filled with particles since $\nabla \cdot \mathbf{u}_p < 0$ there, while flow regions where vortical motion prevails over straining ($Q > 0$) tend to be devoid of particles[19,33]. For charged particles, $St_{k,\pm}^{(el)} > 0$, and the description also depends on the net charge density ($n_+ - n_-$). In particular, in regimes where electrostatic effects prevail, $St_{k,\pm}^{(ae)}/St_{k,\pm}^{(el)} \ll 1$, particle clouds with net negative charge ($n_- > n_+$) drive away negatively charged particles and attract positively charged ones, thus resembling the self-regulating dynamics of the internally generated electric field in canceling preferential concentration, as described above. The opposite trend occurs in positively charged clouds, as schematically shown in Fig. 3.

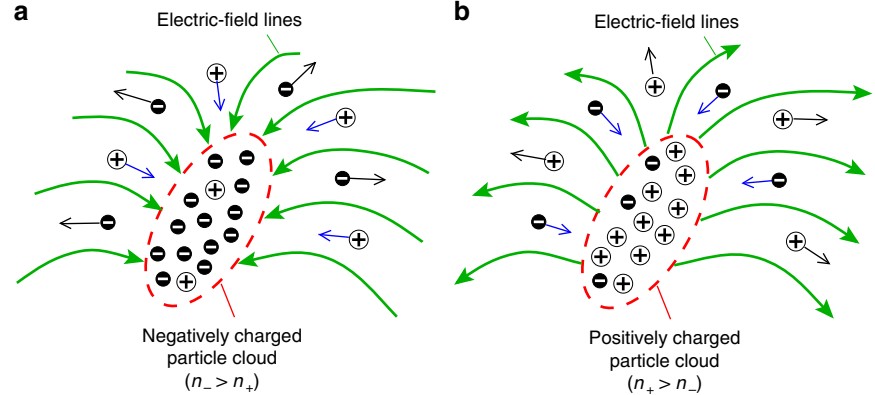

**Fig. 3** Mitigation effects of electrostatics on preferential concentration. **a** A cloud of negatively charged particles generates incoming electric-field lines that steer negative charges away and concentrate positive charges, which tends to electroneutralize the particle charge distribution and overrides the preferential concentration created by the turbulent flow field (schematics, not to scale). **b** A similar but opposite phenomenon is illustrated for a positively charged cloud

In the present simulations, the Stokes-number ratio is $St_{k,+}^{(ae)}/St_{k,+}^{(el)} = St_{k,-}^{(ae)}/St_{k,-}^{(el)} = 10$, but despite the prevailing aerodynamic effects, the internally generated electric fields are sufficiently large to noticeably modify the structure of the particle concentration field. It should be emphasized, however, that Eq. (13) is an asymptotic approximation for small particle inertia and electromigration velocities, with finite values of these quantities entering in the problem to decorrelate the dynamics of the carrier phase from those of the dispersed phase.

**The structure of the electric field**. The accumulation of negatively charged small particles in clouds has the fundamental effect of generating spatially coherent electric fields. This is shown in Fig. 4 by comparing the electric fields generated in regimes where preferential concentration is significant (case #2) or negligible (case #3). In particular, case #3 in Fig. 4b corresponds to charged particles that are ballistic to the small eddies, and is therefore characterized by relatively uniform spatial distributions of both classes of particles. Correspondingly, the number-density fields in case #3 have a very small content of spectral energy at high wavenumbers, as observed in Fig. 2f. The absence of any significant preferential concentration in case #3 leads to spatially incoherent electric fields whose peak intensities are of order $n_0 \ell_k q_+/\epsilon_0$, as observed in the contours in Fig. 4b. The spectral electrostatic energy of this electric field, denoted as $\mathcal{E}_E$ and computed similarly to $\mathcal{E}_n$ by spherically averaging the multiplication of the fast-Fourier transform of $\overline{\mathbf{E}}_{FMM}/\sqrt{2}$ by itself, has a slope close to 2, which resembles white noise, as indicated in Fig. 5a. In contrast, the preferential concentration of negatively charged small particles that dominates case #2 produces stronger electric fields of order $10 n_0 \ell_k q_+/\epsilon_0$, as shown in Fig. 4a. Such electric fields are spatially coherent and their maximum intensities occur in the vicinity of the clouds of negatively charged small particles. Additionally, Fig. 5a shows that the electric fields in case #2 have a much higher spectral energy content at low wavenumbers than in case #2, particularly near the integral wavenumber of the turbulence $(2\pi/\ell)\ell_k = 0.06$, where the spectral electrostatic energy created by the preferential concentration effect is >200 times larger than that at the Kolmogorov scales.

The 10-fold increase of the electric field observed after switching from case #3 to case #2, where the preferential concentration of negatively charged small particles is significant, is particularly evident by the rightward shift in the probability density functions (PDFs) provided in Fig. 5b. In addition, Fig. 5b shows that the effect of decreasing the particle charges $q_+$ and $q_-$ by a factor of 10, which corresponds to a decrease of both electric Stokes numbers by a factor of 10, is to decrease the electric field normalized with the baseline charge level albeit by just a factor of order unity. However, the spectral electrostatic energy content of this diminished electric field was observed in the results to be shifted toward larger scales relative to case #2, because of the tendency of the aforementioned mitigating effect of the electric field on preferential concentration to decrease in intensity as the electric charge decreases. These considerations highlight the fact that regimes at low electric Stokes numbers are more effective at producing coherent electric fields at the large scales of turbulence, although the resulting values are also correspondingly smaller.

As sketched in Fig. 1, the mechanism of aerodynamic generation of electric fields described above rests upon the segregation of charges induced by turbulence. This charge separation is examined in Fig. 5c in terms of the energy spectrum $\mathcal{E}_q$ of the fluctuations of net charge density $(n_+ - n_-)q_+$, with $\mathcal{E}_q$ being computed analogously to $\mathcal{E}_n$. Specifically, the occurrence of preferential concentration in case #2 leads to a significant high-wavenumber enhancement of $\mathcal{E}_q$ because of the long, negatively charged filamentous structures created by turbulence. The wavenumber associated with the peak $\mathcal{E}_q$ is larger than those related to the maxima of $\mathcal{E}_{n,-}$ and $\mathcal{E}_{n,+}$ due to the partial electroneutralization of the clouds by the surrounding positively charged large particles, which makes the structures of charge segregation narrower than the clouds of negatively charged small particles.

The excess of spectral energy of net charge-density fluctuations at high wavenumbers in case #2 leaks into the energy spectra of electric field and electric potential at increasingly larger scales because of the relation $\mathcal{E}_q = \epsilon_0 \kappa \mathcal{E}_E = \epsilon_0 \kappa^2 \mathcal{E}_\phi$ that holds in electrostatics, where $\mathcal{E}_\phi$ is the energy spectrum of the fluctuations of the FMM-derived mesoscopic electric potential $\overline{\phi}_{FMM}$. As a result, the latter becomes preferentially organized in much larger scales than those of the electric field and net charge density. This leads to a total decay in $\mathcal{E}_\phi$ in Fig. 5d of ~6 orders of magnitude along 2 decades of wavenumbers in a similar way to the kinetic-energy spectrum $\mathcal{E}_K$ of the carrier phase in Fig. 5a.

**Atmospheric rarefaction effects**. Whether the range of dimensionless parameters considered above is of relevance for realistic dust storms is a question that cannot be categorically answered due to the large variabilities in flow conditions and particle

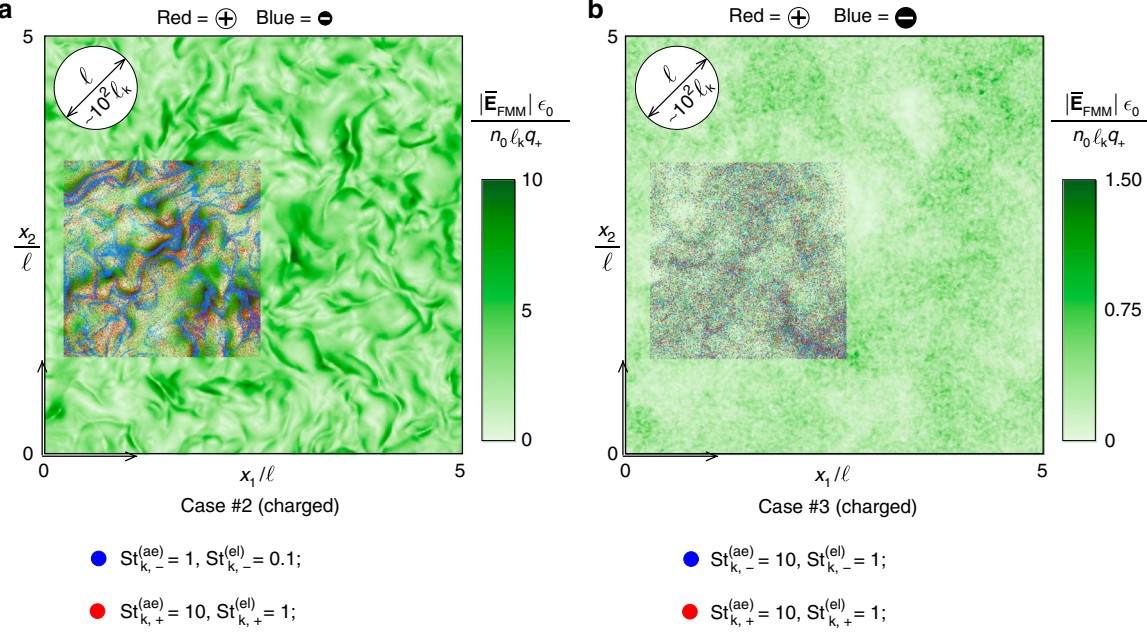

**Fig. 4** Emergence of turbulence-driven electric fields. Instantaneous cross-sectional contours of the FMM-derived mesoscopic electric field $\overline{\mathbf{E}}_{FMM}(\mathbf{x})$ for **a** case #2, in which negatively charged small particles preferentially concentrate, and **b** case #3, in which none of the particle classes preferentially concentrate in any significant manner. The insets show the superposed local spatial distribution of negatively charged small particles (blue color dots) and positively charged large particles (red color dots) in a constant-$x_3$ slice of thickness equal to the Kolmogorov length $\ell_k$. The integral length $\ell$ and its equivalent size in Kolmogorov units (~$100\ell_k$) are provided for convenience in the left upper corners

properties reported in the literature, particularly for extra-terrestrial atmospheres[10,12,15]. However, there are aspects related to the influences of low ambient pressures in favoring the phenomenon studied here that are worth discussing in terms of dimensional quantities of practical interest.

Consider, for instance, two separate turbulent flows in the same homogeneous configuration studied above. Both flows are subject to the same large-scale forcing and, correspondingly, the same root-mean-square velocity fluctuation and integral length, which are chosen here, respectively, as $u_\ell \sim 2 \, \text{m s}^{-1}$ and $\ell \sim 20 \, \text{cm}$ for illustration purposes. The rest of the parameters are $a_+ \sim 40 \, \mu\text{m}$ and $a_- \sim 10 \, \mu\text{m}$, $q_+ = -q_- \sim 50 \, \text{fC}$, $\rho_p \sim 2650 \, \text{kg m}^{-3}$, and $n_0 \sim 5 \times 10^8 \, \text{m}^{-3}$. These values lie within the ranges classically reported in the space-weather literature[1–6,8–17]. However, the gas environments in the two flows considered here are different. One of them emulates the terrestrial atmosphere with air ($\mu_\oplus = 1.8 \times 10^{-5} \, \text{N s m}^{-2}$) at 1 bar and 298 K, thereby resulting in a density $\rho_\oplus = 1.2 \, \text{kg m}^{-3}$. The other one emulates the rarefied, $CO_2$-rich Martian atmosphere ($\mu_{\sigma} = 1.3 \times 10^{-5} \, \text{N s m}^{-2}$) at 6.9 mbar and 210 K, which gives $\rho_{\sigma} = 1.6 \times 10^{-2} \, \text{kg m}^{-3}$. The large disparities in densities, $R = \rho_\oplus/\rho_{\sigma} = 75$, and kinematic viscosities, $\mathcal{V} = \nu_{\sigma}/\nu_\oplus = 54$, have important effects on the relative magnitude of the resulting dimensionless parameters as follows. Perhaps the most significant effect of ambient rarefaction is on the Taylor–Reynolds number, which changes from $\text{Re}_{\lambda,\oplus} \sim 630$ to

$$\text{Re}_{\lambda,\sigma} = \mathcal{V}^{-1/2}\text{Re}_{\lambda,\oplus} \sim 85. \tag{14}$$

As a consequence, the Kolmogorov length and time scales increase from $\ell_{k,\oplus} \sim 96 \, \mu\text{m}$ and $t_{k,\oplus} \sim 0.6 \, \text{ms}$, to $\ell_{k,\sigma} = \mathcal{V}^{3/4}\ell_{k,\oplus} \sim 1.9 \, \text{mm}$ and $t_{k,\sigma} = \mathcal{V}^{1/2}t_{k,\oplus} \sim 4.6 \, \text{ms}$. For the terrestrial environment, the corresponding Stokes numbers are $\text{St}^{(ae)}_{k,-,\oplus} = 5$, $\text{St}^{(ae)}_{k,+,\oplus} = 83$, $\text{St}^{(el)}_{k,-,\oplus} = 0.02$, and $\text{St}^{(el)}_{k,+,\oplus} = 0.01$. In

contrast, ambient rarefaction causes a decrease in the aerodynamic Stokes numbers,

$$\text{St}^{(ae)}_{k,-,\sigma} = \frac{R\text{St}^{(ae)}_{k,-,\oplus}}{\mathcal{V}^{3/2}} \sim 1, \quad \text{St}^{(ae)}_{k,+,\sigma} = \frac{R\text{St}^{(ae)}_{k,+,\oplus}}{\mathcal{V}^{3/2}} \sim 16, \tag{15}$$

along with an increase in the electric Stokes numbers,

$$\text{St}^{(el)}_{k,-,\sigma} = \frac{R\text{St}^{(el)}_{k,-,\oplus}}{\mathcal{V}^{1/2}} \sim 0.3, \quad \text{St}^{(el)}_{k,+,\sigma} = \frac{R\text{St}^{(el)}_{k,+,\oplus}}{\mathcal{V}^{1/2}} \sim 0.1. \tag{16}$$

The mean absolute deviations of the mesoscopic electric fields created by turbulent particle dispersion in the terrestrial and Martian environments are

$$E_\oplus = c_\oplus \frac{n_0\ell_{k,\oplus}q_+}{\epsilon_0} \sim c_\oplus \times 270 \, [\text{V m}^{-1}] \tag{17}$$

and

$$E_{\sigma} = \left(\frac{c_{\sigma}}{c_\oplus}\right)\mathcal{V}^{3/4}E_\oplus = c_{\sigma}\frac{n_0\ell_{k,\sigma}q_+}{\epsilon_0} \sim c_{\sigma} \times 5400 \, [\text{V m}^{-1}], \tag{18}$$

respectively, where $c_\oplus$ and $c_{\sigma}$ are prefactors that have to be computed by numerical integration of each problem.

In principle, since the characteristic Reynolds number of the flow in the terrestrial environment is too high, no grounded guess of $c_\oplus$ can be made in view of the numerical results presented above. However, the associated aerodynamic Stokes numbers suggest only marginal levels of preferential concentration. As a result, large values of order $c_\oplus \gtrsim 10^4$ required for electric breakdown (i.e., $E_\oplus > 3 \, \text{MV m}^{-1}$) are not easily conceptualized based on the fractional electric fields observed in Fig. 5b for case #3 where the particles are mostly ballistic, thereby suggesting that

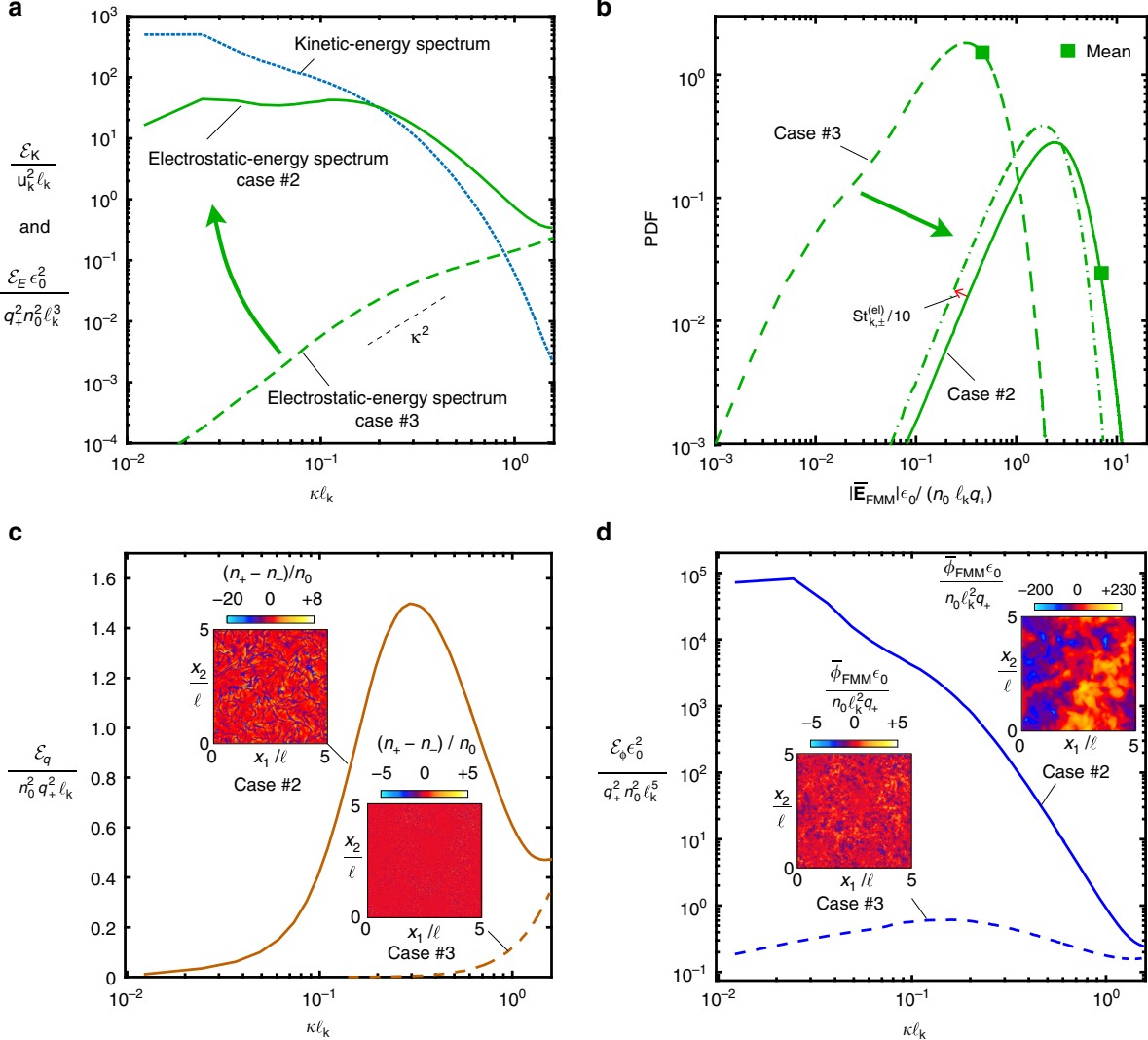

**Fig. 5** Effects of the turbulent dispersion of particles on the electric field. **a** Ensemble-averaged electrostatic and kinetic-energy spectra as a function of the wavenumber. **b** Ensemble-averaged normalized probability density functions (PDF) of the modulus of the FMM-derived mesoscopic electric field (case #2 mean: 7.3, case #2 95th percentile: 13.6; case #3 mean: 0.4, case #3 95th percentile: 0.9). **c** Ensemble-averaged energy spectra of the net charge-density fluctuations as a function of the wavenumber, including instantaneous cross-sectional contours (insets). **d** Ensemble-averaged energy spectra of the electric-potential fluctuations as a function of the wavenumber, including instantaneous cross-sectional contours (insets)

this aerodynamic mechanism is unlikely to produce electric discharges for this set of parameters.

On the other hand, the dimensionless parameters (Eqs. (14), (15), and (16)) corresponding to the Martian environment are similar to the ones of the simulation case #2 studied above, where the negatively charged small particles preferentially concentrate, the positively charged large particles do not preferentially concentrate significantly, and the electric Stokes numbers are sufficiently small to render weak electric interactions, as in Eq. (9). Because of the increase in the Kolmogorov length as the pressure decreases, the characteristic electric field $n_0 \ell_k q_+ / \epsilon_0$ is much larger in the rarefied environment. In particular, the results in Fig. 5b indicate that $c_{\mathcal{E}} \sim 7.3$, thereby making $E_{\mathcal{E}} \sim 40$ kV m$^{-1}$, which, in principle, could trigger electric breakdown in these rarefied conditions (i.e., $E_{\mathcal{E}} > 25$ kV; e.g., see Fig. 1 in ref. [16] for calculation of breakdown values as a function of $CO_2$ number density). In interpreting these estimates, note that the low pressures in the Martian atmosphere cause a significant increase in the mass loading, $\alpha_{\pm,\mathcal{E}} = R\alpha_{\pm,\oplus}$, with $\alpha_{-,\oplus} \sim 0.004$ and $\alpha_{+,\oplus} \sim$

0.2 in this example, which may require consideration of two-way coupling effects that have been neglected in the numerical simulations presented above. This increase in mass loading is, however, associated with very small volume fractions $\phi_{\nu,+} = (4/3)\pi n_0 a_+^3 \sim 10^{-4}$ and $\phi_{\nu,-} = (4/3)\pi n_0 a_-^3 \sim 10^{-6}$, which are independent of the ambient pressure.

## Discussion

This study shows that mesoscopic electric fields can be aerodynamically created in dilute bi-disperse suspensions of oppositely charged particles. The generation mechanism relies on the preferential concentration of the particle class whose response time is similar to the turnover time of the smallest eddies, whereas the particles of the other class are ballistic to the small eddies and do not preferentially concentrate in any significant way.

While the suspension of ballistic particles does not engender any net electric effect, mesoscopic electric fields are generated in suspensions where one of the particle classes preferentially

concentrate. These fields span much larger characteristic lengths than the mean inter-particle spacing, the length scale of the smallest eddies, and the size of the thin regions where charge imbalances occur.

The dynamics involved in the generation of electric fields are self-regulating, in that the preferential concentration of particles decreases when they become electrically charged. This mitigation can be rationalized using the continuum equations of the dispersed phase.

The simulations make use of a FMM in order to drastically reduce the computational cost otherwise incurred by the many-body electrostatic problem. This technique enables a fast computation of the Coulombic forces on the discrete particles while avoiding inherent inaccuracies arising in alternative, homogenized formulations of the electrostatic problem.

The formulation presented in this work has not incorporated several important factors that may warrant further investigations. These include collision-induced charge transfer in denser and more intensely electrified flows, mean wind shear, and turbulence anisotropy effects in flows near the ground, two-way coupling effects and molecular slip on particles in rarefied atmospheres, and discharge chemistry and flow compressibility effects in lightning inception and propagation upon electric breakdown in the carrier gas. In addition, extensions of early theoretical work on preferential concentration[19,33,34] may be worth pursuing for an increased understanding of the phenomenon under electric interactions.

## Methods

**Simulations.** In this study, DNS of incompressible homogeneous-isotropic turbulence laden with point particles are performed in a Cartesian, cubic, triply periodic computational domain of side length $5\ell$ and $256^3$ grid elements. The continuity and linearly forced momentum equations

$$\nabla \cdot \mathbf{u} = 0, \quad \frac{\partial \mathbf{u}}{\partial t} + \mathbf{u} \cdot \nabla \mathbf{u} = -\frac{1}{\rho}\nabla\Pi + \nu\nabla^2\mathbf{u} + A\mathbf{u}, \quad (19)$$

are numerically integrated for the carrier phase, where $\Pi$ is a hydrodynamic pressure computed from the integration of a Poisson equation obtained by taking the divergence of the momentum equation and making use of the divergence-free constraint for the velocity field[35]. The forcing coefficient $A$ is such that statistically steady homogeneous-isotropic turbulence at constant kinetic energy is maintained[36]. The resulting Taylor–Reynolds number of the simulations is $Re_\lambda = (15Re_\ell)^{1/2} = u_\ell\lambda/\nu = 85$, where $\lambda$ is the Taylor microscale. The resolution of the computational grid is $\kappa_{max}\ell_k = 1.6$, where $\kappa_{max} = \pi/\Delta$ is the maximum wavenumber and $\Delta$ is the grid spacing. The initial conditions used for integrating (Eq. (19)) involve a synthetic, solenoidal-isotropic velocity field with a prescribed Passot–Pouquet kinetic-energy model spectrum[36,37].

The formulation of the dispersed phase is based on the Lagrangian description given by Eqs. (1)–(3) using $N \sim 168 \times 10^6$ bi-disperse inertial particles equally repartitioned among the two classes, which warrant a mean number density $n_0 = (N/2)/(5\ell)^3 = (N/2)/(256\Delta)^3$ equivalent to five particles of each class per elementary grid volume $\Delta^3$. The resulting mean inter-particle distance is $\delta_I = (2n_0)^{-1/3} \sim \ell_k$.

Once the flow has reached a statistically steady state, the particles are randomly seeded in kinematic equilibrium with the local flow velocity, which is evaluated at all times at the particle position using a trilinear interpolation. After sufficiently long times compared to $t_{a,\pm}$ have passed, 10 snapshots equally spaced in time are recorded during an interval $10t_\ell$ for ensemble averaging. Equations (1)–(3), and (19) are solved simultaneously using a fourth-order Runge–Kutta method for time advancement and a second-order finite-differences central scheme for the spatial operators[35].

Three different cases (#1, 2, and 3) are computed in the simulations that are characterized by different values of Stokes numbers (Eqs. (6) and (7)). Specifically, case #1 corresponds to uncharged particles for which the small ones (denoted by the subindex $<$) are subject to preferential concentration, while the larger ones (denoted by the subindex $>$) are ballistic to the small eddies, namely

$$St_{k,<}^{(ae)} = 1, \text{ and } St_{k,>}^{(ae)} = 10, \quad (20)$$

with zero electric Stokes numbers for both classes. Case #2 considers charged particles where the small (negatively charged) ones preferential concentrate, whereas the large (positively charged) ones remain comparatively uniformly

distributed in space,

$$St_{k,-}^{(ae)} = 1, \ St_{k,-}^{(el)} = 0.1, \ St_{k,+}^{(ae)} = 10, \text{ and } St_{k,+}^{(el)} = 1. \quad (21)$$

In contrast, case #3 corresponds to charged particles not subject to any significant preferential concentration,

$$St_{k,-}^{(ae)} = St_{k,+}^{(ae)} = 10, \text{ and } St_{k,-}^{(el)} = St_{k,+}^{(el)} = 1. \quad (22)$$

The choice of dimensionless parameters (Eqs. (21) and (22)) has been made for illustration purposes to yield equal Stokes-number ratios $St_{k,+}^{(ae)}/St_{k,+}^{(el)} = St_{k,-}^{(ae)}/St_{k,-}^{(el)} = 10$ in both simulation cases. This enables isolation of effects related to preferential concentration by keeping constant $St_{k,+}^{(ae)}$ and $St_{k,+}^{(ae)}$ across cases #2 and #3, while inducing significant preferential concentration of the negative particles by varying $St_{k,-}^{(ae)}$ from 10 in case #3, to 1 in case #2. A noteworthy peculiarity of the selection (Eqs. (21) and (22)) is the change of $St_{k,-}^{(el)}$ from 1 in case #3, to 0.1 in case #2, which keeps the relative strength of hydrodynamic and electric effects on the particle clouds (i.e., the first and second terms on the right-hand side of Eq. (13)) the same in both particle classes and in both simulation cases. Selection of smaller values of $St_{k,-}^{(el)}$ in case #2, with $St_{k,+}^{(el)} < St_{k,-}^{(el)}$ to reflect the larger size of the positive particles in dimensional applications (assuming that all other dimensional parameters are exactly equal for both classes), does not lead to significant differences in the overall solution since $St_{k,+}^{(el)} = 1$ is already below the threshold for weak electric interactions specified in Eq. (9).

**Computation of the electric force on the particles.** In principle, in Eq. (1), the electric force on a particle can be computed in different ways depending on whether the dispersed phase is assumed to be a continuum, or alternatively, is more realistically considered as a collection of discrete particles. In the former case, the electric force $\mathbf{F}_p$ would be computed by multiplying the particle charge $q_p$ by a homogenized electrostatic field $\overline{\mathbf{E}} = -\nabla\overline{\phi}$ obtained as a solution of the Gauss (Eq. (4)) written in terms of the corresponding potential $\overline{\phi}$, namely

$$\nabla^2\overline{\phi} = -(n_+ - n_-)q_+/\epsilon_0, \quad (23)$$

which is subject to triply periodic boundary conditions. In Eq. (23), the number densities $n_+$ and $n_-$ are continuum representations of the particle concentration field defined on the DNS grid $\mathbf{x}$ and which hereafter are obtained by box-counting the particles with a nearest-neighbor approach[38]. The homogenization involved in computing (23) implies that $\overline{\mathbf{E}}(\mathbf{x})$ is the local electric field averaged over a sufficiently large number of particles, whose diameters and inter-particle distances are much smaller than the homogenization length $\Delta$. Although this approach is consistent with Eulerian formulations of the dispersed phase, it leads to inaccurate computations of the electric force in Lagrangian formulations as Eqs. (1)–(3) subject to a finite number of particles. To understand this, note that turbulence tends to break down the uniformity in the spatial distribution of particles by generating highly dense and highly devoid regions (see Results section), in a way that the local number of particles per cell oscillates in space from zero to $O(100)$ in the present simulations, and consequently $\delta_I$ fluctuates from approximately $0.1\Delta$ to distances spanning multiple grid cells. Consider, for instance, the case where two particles of opposite charge are present in a grid cell thereby yielding zero electric charge inside, and consequently, zero mesoscopic electric field because of the Gauss theorem. Since in practical implementations of this approach into Eq. (1), the force on the particle requires differentiation of the potential $\overline{\phi}$ and interpolation of the mesoscopic field $\overline{\mathbf{E}}$ onto the particle position $\mathbf{x}_p$, the calculation would yield zero electric force on both particles even though the electric force computed directly from the Coulomb theory clearly pushes them toward each other.

A direct calculation of the electric force created by all other particles on the $p$-th particle using Coulomb's point-charge expression

$$\mathbf{F}_p = \sum_{j=1, j\neq p}^{N} \frac{q_p q_j}{4\pi\epsilon_0} \frac{\mathbf{x}_j - \mathbf{x}_p}{|\mathbf{x}_j - \mathbf{x}_p|^3} \quad (24)$$

therefore becomes more convenient in systems laden with a finite number of particles. Note however that this operation results in oftentimes untenable CPU-time requirements of order $N^2$ per time step. This hindrance can be circumvented by methods for approximating the far-field electric potential while employing Eq. (24) for particles sufficiently close to $\mathbf{x}_p$, as in the FMM approach proposed in ref. [26] and followed in the present work. This method, which is briefly described below, results in a significant reduction of computational cost by a factor of order $N/\log N$, which in practice is of order $N$ since $N \gg 1$. In addition, FMM does not require to solve Eq. (23) since it does not make use of the continuum assumption as opposed to earlier work[28], it does not involve tunable distances to blend continuum and discrete approaches as in other contemporary methods such as the particle–particle/particle–mesh ($P^3M$) approach[29], and it does not employ tunable cutoff distances utilized in recent studies[30] beyond which the effects of distant particles are neglected.

Consider a particle p located at large distances $r$ from a cloud of characteristic size $L_c$, in the sense that $r/L_c \gg 1$. The exact electric potential at the particle location

$$\phi(\mathbf{x}_p) = \sum_{j=1}^{n_c} \frac{q_j}{4\pi\epsilon_0 |\mathbf{x}_j - \mathbf{x}_p|} \tag{25}$$

can be approximated using the Laplace expansion

$$\phi(\mathbf{x}_p) \simeq \frac{1}{4\pi\epsilon_0} \sum_{l=0}^{\infty} \sum_{m=-l}^{l} (-1)^m \mathcal{I}_l^{-m}(\mathbf{x}_p) \sum_{j=1}^{n_c} q_j \mathcal{R}_l^m(\mathbf{x}_j) \tag{26}$$

in terms of the regular and irregular solid harmonics $\mathcal{R}_l^m$ and $\mathcal{I}_l^m$, respectively, where $n_c$ is the number of particles in the cloud[39]. In Eq. (26), the first term in the expansion ($l=0$) corresponds to $\phi(\mathbf{x}_p) = \sum_{j=1}^{n_c} q_j/(4\pi\epsilon_0 |\mathbf{x}_p|)$, which represents the effect at $\mathbf{x}_p$ of a point charge of magnitude equal to the cloud charge, while the second ($l=1$) and third ($l=2$) terms vary quadratically and triadically with the inverse of the distance, respectively, and represent the dipole and quadrupole effects of the cloud at $\mathbf{x}_p$. The contribution of each term in Eq. (26) diminishes as $l$ increases.

In the FMM, the particle clouds are identified at every Runge–Kutta time substep using an octree subdivision of the computational domain whereby a recursive partition into 8 cubic subdomains is performed until each of them (the octree leaves) does not contain more than a maximum number of particles, which here is chosen to be $N_{c,max} = 128$. Subsequently, the Laplace expansion (Eq. (26)) is performed to fifth order ($l=5$). Periodicity is imposed by duplicating the instantaneous spatial distribution of particles twice on each direction. The total potential created by clouds of particles surrounding $\mathbf{x}_p$ is computed exactly using Eq. (25) if they are contained in leaves that have at least one node shared with the leaf that contains $\mathbf{x}_p$, while the expansion (Eq. (26)) particularized to fifth order is used to compute the long-range effects of clouds in leaves beyond the adjacent ones. Lastly, the FMM-derived electric force (Eq. (2)) is computed based on an electrostatic field $\mathbf{E}_p = -\nabla\phi|_{\mathbf{x}=\mathbf{x}_p}$ resulting from the analytical differentiation of the linear superposition of all the aforementioned contributions to the potential created by all the other particles at $\mathbf{x}_p$. Note that the differentiation of the short-range component of the potential (Eq. (25)) leads to the exact electrostatic force (Eq. (24)). As a result, a limiter for the minimum inter-particle distance is required in order for the force not to become unbounded. This limiter is set to $\ell_k/100$ and does not have any significant consequence in the regimes analyzed here, where the suspensions are dilute and the electric Stokes numbers are relatively small.

Quantifications of collision rates were made in a first approximation by studying the frequency with which the inter-particle distance limiter was triggered during one integral time in the simulations. The value of the volume fraction associated with the cross section imposed by the limiter is $\phi_v \sim 10^{-5}$. The results suggested that collisions played a secondary role in the regimes investigated here and therefore were excluded from the model. For instance, in case #2, a fraction of order 0.04% of the total number of particles involved opposite-sign charged particles approaching each other at distances smaller than the limiter's value. These events were characterized as necessary but not sufficient conditions for collisions. The proportion of collisions leading to charge transfer between particles is smaller and typically amounts to 20% of the fraction of collided particles with opposite signs[9]. As a result, ~0.01% of the total amount of particles could have potentially been subjected during one integral time to charge transfer, thereby requiring a total integration time of order 10,000 integral times for observing any significant effect related to this phenomenon, which in the numerical examples discussed above would have translated into an exceedingly long period of 17 min of real time that has no physical relevance for the questions addressed here.

In this study, the FMM is implemented via the open-source library ExaFMM[40] for massively parallel environments. Using this framework, cases #2 and 3 with charged particles involved 250,000 CPU hours on 4176 cores (Intel Xeon E5-2695) in the Lawrence Livermore National Laboratory (LLNL) Quartz supercomputer, whereas the uncharged case #1 required 60,000 CPU hrs on 1024 cores in the same machine.

**FMM-derived mesoscopic electric field**. In order to characterize the statistics of the electric fields collectively generated by the charged particles, the results presented above make use of the FMM-derived mesoscopic electric field $\overline{\mathbf{E}}_{FMM}(\mathbf{x})$ obtained by numerically differentiating the FMM-derived mesoscopic potential $\overline{\phi}_{FMM}(\mathbf{x})$ at every point on the DNS grid. Specifically, $\overline{\phi}_{FMM}(\mathbf{x})$ is computed by integrating in a post-processing step the Gauss law (23), where the number densities $n_\pm$ are calculated from particle positions obtained by integrating Eqs. (1) and (3) with $\mathbf{F}_p$ being the FMM-derived electrostatic force described above. Note that the projection of the full FMM electric field on the DNS grid resulting from analytically differentiating the short- and long-range components of the full FMM potential $\phi(\mathbf{x}_p)$, namely Eqs. (25) and (26) particularized for $\mathbf{x}_p = \mathbf{x}$, becomes singular when a particle position $\mathbf{x}_j$ is close to a grid node, and is therefore a field that is not further analyzed here since its high-wavenumber behavior is highly grid-dependent. In practice, $\overline{\mathbf{E}}_{FMM}(\mathbf{x})$ behaves as

a low-pass filtered quantity whose spectral response closely reproduces that of the full FMM electric field up to very high wavenumbers where it decays faster because it does not contain any singularities related to particles overlapping grid nodes.

**Data availability**. The data that support the plots and other findings of this study are available from the authors upon request.

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

## Acknowledgements

This investigation was funded by the Advanced Simulation and Computing (ASC) program of the US Department of Energy's National Nuclear Security Administration (NNSA) via the PSAAP-II Center at Stanford University, grant #DE-NA0002373. This work was performed during the visit of the first author to the Stanford University Center for Turbulence Research in Fall 2016 and Winter 2017.

## Author contributions

M.D.R. and J.U. designed the numerical experiments. M.D.R. and J.U. analyzed the data and developed the theoretical model. M.D.R. and J.U. wrote the paper.

## Additional information

**Competing interests:** The authors declare no competing interests.

