## [Peer Review File · Nature Communications]

Reviewer #1 (Remarks to the Author):

The authors numerically investigate separation of positively and negatively charged grains in a turbulent gas. The main result of the paper is preferential concentration of negatively charged smaller grains surrounded by uniformly distributed positive particles.

The differential charging of grains of different sizes in a neutral atmosphere, known to occur due to the triboelectric effect, is caused by multiple mutual collisions between grains (see e.g. refs. 8 and 11 of the manuscript). However, the authors completely neglect collisions in their simulations, i.e. they assume that collisions generate charges on grains and then mysteriously disappear! Of course, one can say that charges may be created by some unknown mechanisms, but then the presented results have little relevance to the Martian atmosphere or similar environments.

Inclusion of collisions in the simulations will probably result in various new effects, such as the charge redistribution and recombination as well as the dust coagulation, all stimulated by the mutual attraction of the oppositely charged grains at the edges of negative clouds. Without such analysis I consider the presented results as absolutely unreliable. Also, I believe that the topic of the paper is more appropriate for specialized journals, e.g. JGR, GRL etc.

The paper should be rejected.

Reviewer #2 (Remarks to the Author):

This paper models the size separation of charged particles embedded in turbulent flow. As triboelectric charging tends to produce particles of opposite polarity based on their size, the charge imbalance caused by size-segregation of the grains can lead to large electric fields. The generated electric field may be large enough to exceed the breakdown barrier in Mars' atmosphere under certain conditions. The results are of interest in designing equipment which will be exposed to dust storms on the Martian surface.

While other published reports have examined the electrification of dust storms in the Martian atmosphere, this paper is novel in that it examines the segregation of small particles in the interstices of vortices in turbulent flow. As such, it can be applied to other systems where turbulent flow occurs in a rarified gas, such as a protostellar clouds or protoplanetary disks.

The modeling and subsequent analysis of the results are convincing to show that significant electric fields develop in the turbulent flow and that it is feasible to cause electric breakdown in the Martian environment. The description of the model is presented in enough detail to reproduce the results. It would be better to give the exact values chosen for the variables in the simulation, instead of giving approximations. For example, I read $a^+ \sim 40$ microns as $a^+ = 40$ microns in the simulation.

However, the explanation of the mechanism of generation of electric fields seems to contain a contradiction. The aerodynamic Stokes number (Eq 6) is defined to be proportional to the radius squared, while the electric Stokes number (Eq. 7), is inversely proportional to the particle radius. Since it is assumed that the small particles are negative, this implies that $St_{el}(k,+)^{ae} \gg St_{el}(k,-)^{ae}$ and that $St_{el}(k,+)^{el} < St_{el}(k,-)^{el}$ ($u_{el,+} < u_{el,-}$). Thus the condition stated for the aerodynamic Stokes numbers is satisfied (Eq . 8), but the condition for the electric Stokes numbers (Eq 9), that the Stokes number for the positive particles is larger than that for the negative particles (and relatedly, the claim that the electromigration velocity for the negative particles is smaller than that for positive particles) cannot be satisfied. Perhaps the inequalities have been switched? The values given for the electric Stokes numbers for the simulations (Eq. 16 and the text above) do correctly give larger values for the negative particles, although I would not expect the electric Stokes numbers to differ by a factor of ~ 2 , given that the radii differ by a factor

of ~ 4 . This discrepancy may be addressed if the exact values are given for each variable, as noted above.

Lorin Matthews

Reviewer #3 (Remarks to the Author):

This manuscript presents a scaling analysis, informed by a detailed computational investigation, on the role of particle-turbulence interactions on self-induced electric fields. Simulations of homogeneous isotropic turbulence laden with bi-disperse particles are conducted for three main cases: (i) uncharged and bidisperse, (ii) oppositely-charged and bidisperse, and (iii) oppositely-charged and monodisperse. Each particle contains a single charge that is either positive or negative. Particles of order unity Stokes number preferentially concentrate in high-strain regions of the turbulent flow, while higher Stokes number particles remain relatively uniformly distributed owing to their ballistic behavior. Because smaller particles contain an opposite charge, this segregation in particle size results in the production of long-wavelength electric fields in the carrier gas — much larger than the mean inter particle scaling or size of the smallest eddies.

The physical scenario is very intriguing, and the analysis presents new insight into potential mechanisms responsible for atmospheric electrification. After reading this paper in detail I believe the scaling analyses and corresponding numerical approach are sound. My only concern is whether the DNS is truly representative of conditions relevant to Martian atmospheres, as this was a main focus of the study. In a rarefied atmosphere, what does Stokes ~ 1 represent? What density ratios and particle sizes would be needed to achieve this? However, the authors correctly point out the shortcomings of the approach: “Whether the range of dimensionless parameters considered above is of relevance for realistic dust storms is a question that cannot be categorically answered due to the large variabilities in flow conditions and particle properties reported in the literature, particularly for extra-terrestrial atmospheres.” Moreover, I agree with their discussion later on that a point charge assumption may not be appropriate. However, I believe the authors adequately acknowledge this and the results reported here are worthy of publication. I only have a few remarks:

1. Despite the low number densities considered, particle accumulation due to preferential concentration in addition to Coulomb attraction may lead to a regime where particle collisions can no longer be ignored. Therefore, simply clipping the inter particle distance as is described in the methods section would miss key physics, even if the results are not sensitive to the value of the clipping distance. What are the authors’ thoughts regarding this?

2. It is stated that aerodynamic forces will always dominate over Coulomb interactions, regardless of the electric Stokes number. I find this surprising, since in the limit of very large charge, I would think the second term on the right-hand side of (1) would control the particle dynamics. As likely charged particles accumulate between eddies (as shown in Fig. 1), wouldn’t Coulomb repulsion act to mitigate this clustering?

NCOMMS-17-34180 - "Aerodynamic generation of electric fields in turbulence laden with charged inertial particles" by Di Renzo & Urzay

Replies to Referee #1

We thank this Referee for the comments, which have been helpful for improving the manuscript (see revisions in **red color**). For convenience, we have taken the liberty of enumerating the Referee's comments.

Referee #1: 1. *The authors numerically investigate separation of positively and negatively charged grains in a turbulent gas. The main result of the paper is preferential concentration of negatively charged smaller grains surrounded by uniformly distributed positive particles. The differential charging of grains of different sizes in a neutral atmosphere, known to occur due to the triboelectric effect, is caused by multiple mutual collisions between grains (see e.g. refs. 8 and 11 of the manuscript).*

Authors: We agree with the Referee that charged particles are most likely created by collisions. This was already stated in the original manuscript in page 1, paragraph 1, line 7: *"This phenomenon involves sand or dust particles, which, upon being lifted off by the wind, become triboelectrically charged by collisions near the ground, where the particle density is large, with small and large particles tending to be negatively and positively charged, respectively, as a result of differential transfer of free electrons from collisions between particles of different sizes [7-9]."*

Referee #1: 2. *However, the authors completely neglect collisions in their simulations, i.e. they assume that collisions generate charges on grains and then mysteriously disappear! Of course, one can say that charges may be created by some unknown mechanisms but then the presented results have little relevance to the Martian atmosphere or similar environments.*

Authors: We did not assume anywhere in the paper that collisions disappear "mysteriously", nor we did say anywhere in the paper that charges "are created by unknown mechanisms", as stated by the Referee. Nonetheless, we have made revisions in the text motivated by the fair comment made by the Referee in order to make sure that we are not projecting that erroneous impression onto the reader (see red color corrections on pages 1 and 2). In addition to those modifications, the following qualitative explanation may address the concern raised by the Referee regarding the exclusion of collisions in the present study (further details, including quantitative assessments of this approximation based on numerical results that justify the exclusion of collisions in our paper, are also provided below in our response to the Referee's comment #3). We apologize in advance for the long reply, which we have summarized in a much shorter manner in the revised manuscript since we trust the reader can read the highlighted references for more details, but in the context of this rebuttal we believe that our point is perhaps best understood this way.

For collisions to have a significant effect on the dynamics, the number density of particles has to be sufficiently high to render a mean free path much smaller than the characteristic length of the problem under investigation. This, however, is not the case in the phenomenon in which we are particularly interested. The justification for this statement is as follows and is also complemented with our quantitative response to Referee's comment #3. High number densities are obviously attained close

to the ground, in the saltation layer, where the grains are closely packed and from where the grains are aerodynamically lifted by the wind shear [e.g., see page 49 in *Physics and Modelling of Wind Erosion*, by Y. P. Shao (Kluwer Academic, Dordrecht, 2000)]. There the grains tumble, roll and collide with each other and with the ground, and as a result they become triboelectrically charged [e.g., see the following text extracted from page 21 in Ref. [5] J. Murphy et al., Field Measurements of Terrestrial and Martian Dust Devils, *Space Sci. Rev.* 203, 39-87 (2016): “[During saltation] they [the grains] jump over the surface, where they reimpact and initiate the motion of particles of a wide range of sizes, including dust. [...] The collisions among particles during saltation are also responsible for electric charge transfer between grains. [...] Considering that, in general, the smallest particles are transported higher into the atmosphere by local turbulence while larger particles remain closer to the surface, this translates in a charge separation and consequently in an enhancement of the atmospheric electric field.”]. We stated something similar in the original text (see our response to the Referee’s comment #1).

It is therefore important to note that collisions close to the ground serve to pre-charge the distribution of particles, and that the spatial segregation of charges must necessarily rely on aerodynamic forces in order to generate mesoscopic electric fields because collisions themselves do not segregate particles. This was stressed in the last sentence of the second paragraph on page 2 in the original manuscript since the main goal of our study is to investigate the segregation of particles by turbulence and the subsequent generation of long-wavelength electric fields (e.g., see paragraphs 2 and 3 in the Introduction section).

As stated in the original text (paragraph 2 in the Introduction), and as also pointed out by the Referee, the problem of charge transfer due to collisions has been studied in early work (e.g., see Ref. [8] in the original manuscript and also others that we did not cite because of space constraints, for instance: Fabian et al., Measurements of electrical discharges in Martian regolith simulant, *IEEE Trans. Plasma Sci.* 29, 2001; Pähtz et al., Why do particle clouds generate electric charges?, *Nature Physics* 6, 2010). However, in these studies, the number density of particles is quite large, and in many cases (e.g., Ref. [8] and Pähtz et al. 2010), the flow of particles is an aerated fountain-like one emerging from a close packed layer of particles, such that the particles necessarily collide back when falling on to the packed layer. Other experiments (e.g., Fabian et al. 2001) are based on agitation of dust layers using mechanical means, for instance by filling a jar with a packed layer of particles in terrestrial gravity and utilizing a a rod for agitation. According to the parameters specified in Fabian et al. 2001, if the particles were to hypothetically fill up the entire jar by perfectly dispersing them with the rod, the number density would have been, at least, of order $10^{11} - 10^{12}$ particles/m³, which would have become locally a much larger value if the dispersion was not perfect due to gravitational stratification and close packing in the layer, as expected in realistic conditions. As a result, in most of these studies collisions become forcefully frequent, as is expected to occur in the saltation layer in the real problem of wind-blown sand or dust.

However, the focus of our study is not on the saltation layer, but above it, where the particles are airborne, the number density of particles is smaller so as to make collisions less frequent, and the wind shear is smaller so as to make the flow more homogeneous and isotropic-like (see also our response to the Referee’s comment #3 for quantitative assessments of collision rates). This focus has been reinforced in the revised text in order (see red color corrections on page 2). The saltation layer on Earth is typically centimeters tall, while the Martian one, is just of order 1 to 6 meters according to RANS numerical simulations by Almeida et al. *PNAS* 105 (2008). Furthermore, note that even within the saltation layer, important studies have neglected midair collisions between the

grains in classic works [e.g. “*The effects of turbulence and midair collisions on particle trajectories are neglected because these effects are relatively small for typical shear velocities*”, text quoted from Kok & Renno, Electrostatics in wind-blown sand, *Phys. Rev. Lett.* 100, (2008)]. The rapid (exponential) decrease in number density above the saltation layer, which makes collisions even less frequent, has been also reported by many studies, including classic ones [e.g., see Fig. 20 in page 63 in *Physics of Blown Sand and Desert Dunes* by R.A. Bagnold (Methuen & Co 1953)].

As a result, based on a number of evidences found in the literature, and in absence of quantitative evidence based on direct experimental measurements of high rates of mid-air collisions in real atmospheric conditions (i.e., field measurements, most particularly in the Martian atmosphere), we believe that mid-air collisions between dust grains above the ground are most likely a second-order effect for the purposes of our study.

Referee #1: *3. Inclusion of collisions in the simulations will probably result in various new effects, such as the charge redistribution and recombination as well as the dust coagulation, all stimulated by the mutual attraction of the oppositely charged grains at the edges of negative clouds.*

Authors: We agree with the Referee that we did not provide appropriate explanations regarding the relevance of collisions in the original manuscript, which perhaps has motivated the fair criticism made by the Referee. We have revised the manuscript in this regard, as detailed below. However, we are convinced that the inclusion of collisions does not result in any new effects of significance for the purposes of this study. Before proceeding further, we would like to emphasize that this criticism made by the Referee has been useful for improving the readability of the manuscript.

From a quantitative standpoint, we believe collisions are not relevant in our simulations. The justification of this statement is based on the following observations. We quantified collision rates in a first approximation by studying the frequency with which the inter-particle distance limiter was triggered during the simulations. The results have been included on page 10 of the revised manuscript (see red color revisions there). When the limiter was triggered, the code returned a flag indicating that two particles were approaching each other at distances smaller than $\ell_k/100$. We characterized these events as necessary but not sufficient conditions for collisions. The value of the volume fraction associated with the cross section imposed by the limiter is $\phi \sim 10^{-5}$ (see also estimates of this quantity included in the examples on page 8 in the revised version). Note that this quantification exercise makes sense for particles satisfying $a_{\pm}/\ell_k \ll 1$ with a_{\pm} the particle radius. However, it should be emphasized that the limit $a_{\pm}/\ell_k \ll 1$ must necessarily be enforced in the formulation even if we were to hypothetically include collisions since the point-particle assumption requires this limit to hold in order to enable the integration of the 2nd Newton’s law of motion for the particles. In addition, the approximation $a_{\pm}/\ell_k \ll 1$ becomes increasingly more realistic as the pressure decreases since the Kolmogorov scale is larger, and as a result $2a_{\pm}/\ell_k$ rapidly attains values of order 100 in rarefied atmospheres, as in the numerical example analyzed on page 8 in the context of dust in Martian atmospheric conditions.

Figure 1 shown below (see next page of this rebuttal letter) provides the cumulative fraction of particles that have collided according to the criterion described above as a function time normalized with the integral time t_{ℓ} (i.e., a single decorrelation time of the large-scale eddies). In all cases, the fraction of particles of opposite signs involved in collision events is extremely small. For instance, for case #2 in our manuscript, only 0.04% of the total number of particles during one integral time

Figure 1: Plot shown as evidence for our response to Referee’s comment #3.

involved opposite-sign charged particles approaching each other at distances smaller than the limiter’s value. This fraction becomes even smaller according to charge transfer models, which typically state that only a small fraction of the collisions lead to charge transfer. For instance, according to Forward et al. Geophys. Res. Lett. (2009), only $(7/32) \times 0.04\% \approx 0.01\%$ of the total amount of particles would have undergone some amount of charge transfer from collisions in case #2 during one integral time. One would have to integrate this problem for more than 10,000 integral times in order to observe significant effects related to the charge transfer by collisions, which in the examples addressed in the last section of the paper (page 8 of the original manuscript) would have translated into 17 minutes of real time. This is a very long time scale that has no physical relevance for the purposes of our study. It is also worth stressing that, as shown in Figure 1 below, preferential concentration actually reduces the collision rate between oppositely charged particles by a factor of 1/4 because of the spatial segregation of particles induced by turbulence, while collisions between particles of the same sign are also suppressed due to Coloumbic repulsion, which makes collisions less relevant altogether.

In light of our replies to comments #2 and #3 made by the Referee, we are reluctant to include in our paper additional effects that (a) are not essential to the conclusions, (b) play no significant role in the simulation cases, (c) would require introduction of additional parameters such as charge transfer rates whose uncertainties are large for both terrestrial and extraterrestrial environments, most particularly in the latter because there are no in-situ experiments as described in the Introduction section, (d) would significantly increase the length of the paper, and (e) would place the flow we want to study in a regime we are not interested in. We believe our concerns about the Referee’s suggestion of significantly modifying and extending our study to include the proposed additional effects are in direct accord with first scientific principles of approaching science with simplicity, in that additional phenomena should not be included without necessity.

Referee #1: 4. *Without such analysis I consider the presented results as absolutely unreliable. Also, I believe that the topic of the paper is more appropriate for specialized journals, e.g. JGR, GRL etc. The paper should be rejected.*

Authors: We refer the Referee to our responses to the comments above. We would also like to emphasize that investigations connected with the same topic treated in this paper have appeared regularly in non-specialized journals such as PNAS, Science, and Nature; see for instance the following papers:

- R. H. Cranfield. Atmospheric electricity during sand storms. *Science* **69**, 474-475 (1929).
- R. Anderson et al. Electricity in volcanic clouds. *Science* **148**, 1179-1189 (1965).
- H. F. Eden and B. Vonnegut. Electrical breakdown caused by dust motion in low-pressure atmospheres: Considerations for Mars. *Science* **180**, 39-87 (1973).
- Almeida et al. Giant saltation in Mars. *PNAS* **105**, 6222-6228 (2008).
- Pätz et al., Why do particle clouds generate electric charges?, *Nature Physics* **6** (2010),

Additionally, the topic has also been highlighted in recent non-specialized TV-series such as *MARS* (Episode 3, National Geographic, 2016. URL: https://www.youtube.com/watch?v=JKBk_Kfucs4&ab_channel=NationalGeographic), thereby suggesting that the topic is of broad interest to the public.

NCOMMS-17-34180 - "Aerodynamic generation of electric fields in turbulence laden with charged inertial particles" by Di Renzo & Urzay

Replies to Referee #2

We thank this Referee for the comments, which have been helpful for improving the manuscript (see revisions in **blue color**). For convenience, we have taken the liberty of enumerating the Referee's comments.

Referee #2: 1. *This paper models the size separation of charged particles embedded in turbulent flow. As triboelectric charging tends to produce particles of opposite polarity based on their size, the charge imbalance caused by size-segregation of the grains can lead to large electric fields. The generated electric field may be large enough to exceed the breakdown barrier in Mars' atmosphere under certain conditions. The results are of interest in designing equipment which will be exposed to dust storms on the Martian surface.*

While other published reports have examined the electrification of dust storms in the Martian atmosphere, this paper is novel in that it examines the segregation of small particles in the interstices of vortices in turbulent flow. As such, it can be applied to other systems where turbulent flow occurs in a rarified gas, such as a protostellar clouds or protoplanetary disks.

The modeling and subsequent analysis of the results are convincing to show that significant electric fields develop in the turbulent flow and that it is feasible to cause electric breakdown in the Martian environment. The description of the model is presented in enough detail to reproduce the results.

Authors: We appreciate the positive comments made by the Referee.

Referee #2: 2. *It would be better to give the exact values chosen for the variables in the simulation, instead of giving approximations. For example, I read $a_+ \sim 40$ microns as $a_+ = 40$ microns in the simulation.*

Authors: We appreciate the criticism made by the Referee. However we would prefer not to provide particular dimensional values within the context of the simulation cases because of the following reasons. We believe fluid mechanics is best understood (and best taught) in the dimensionless world of parameters. The conservation equations integrated in this study only depend on four non-dimensional parameters: the Reynolds number Re_ℓ (or equivalently, the Taylor Reynolds number Re_λ), and the four Stokes numbers $St_{k,+}^{(ae)}$, $St_{k,-}^{(ae)}$, $St_{k,+}^{(el)}$ and $St_{k,-}^{(el)}$. These parameters are defined on pages 2 and 3 of the original manuscript, and the physical character of the solution is entirely determined by their values irrespectively of the choice of dimensional parameters. By developing our discussion through dimensionless numbers we provide an increased universality to the solutions that would be lost by committing ourselves to a particular set of parameters out the many sets that would lead to the same dimensionless parameters, and which would have probably drawn heavy criticism since a canonical set of dimensional parameters for dust storms does not exist (particularly for Mars). In integrating the equations, we selected values for those five dimensionless parameters, whose orders of magnitude we predicted would be physically relevant based on typical flow conditions. We believe this is the most elegant and general way of introducing the problem on page 3 and presenting the results until page 7, rather than pre-tuning particular dimensional parameters whose values are

uncertain and variable in the literature. However, a fair question that could be also asked in light of this discussion is whether our choices for the dimensionless parameters were indeed relevant, but we believe we tackled this question in the last subsection of the Results section (page 8) by noting that previously reported orders of magnitude for dimensional parameters tend to render dimensionless parameters in the same ballpark as the ones we are employing in the simulations (with one notable exceptions concerning the exceedingly high Reynolds numbers found in the terrestrial environment, which represents an everlasting computational limitation we always encounter when performing Direct Numerical Simulations of turbulent flows). In reporting these dimensional parameters on page 8, we believe we should not replace approximate signs by equal signs, since these values are not exact in the literature. Lastly, in an exercise of scientific honesty, we emphasize that the entire result of the conundrum addressed in this study cannot be but an order-of-magnitude estimate when applied to the real world scenario of a Martian dust storm. We have stressed this point throughout the manuscript by using approximate signs except when exact data was required to warrant repeatability of the simulations. Hopefully in the future we will enjoy much more computational power and richer experimental data for studying dust storms including direct measurements by Mars rovers or surface exploration crews, but for now we believe the implications raised by the results presented in our manuscript can motivate a long list of interesting new fundamental questions, including for instance whether this canonical electrification mechanism enabled by preferential concentration is embedded within the turbulent flow core of dust devils.

Referee #2: *3. However, the explanation of the mechanism of generation of electric fields seems to contain a contradiction. The aerodynamic Stokes number (Eq 6) is defined to be proportional to the radius squared, while the electric Stokes number (Eq. 7), is inversely proportional to the particle radius. Since it is assumed that the small particles are negative, this implies that $St_{k,+}^{(ae)} \gg St_{k,-}^{(ae)}$ and that $St_{k,+}^{(el)} < St_{k,-}^{(el)}$ ($u_{el,+} < u_{el,-}$). Thus the condition stated for the aerodynamic Stokes numbers is satisfied (Eq. 8), but the condition for the electric Stokes numbers (Eq 9), that the Stokes number for the positive particles is larger than that for the negative particles (and relatedly, the claim that the electromigration velocity for the negative particles is smaller than that for positive particles) cannot be satisfied. Perhaps the inequalities have been switched? The values given for the electric Stokes numbers for the simulations (Eq. 16 and the text above) do correctly give larger values for the negative particles, although I would not expect the electric Stokes numbers to differ by a factor of 2, given that the radii differ by a factor of 4. This discrepancy may be addressed if the exact values are given for each variable, as noted above.*

Authors: To be completely frank, we actually agree with the Referee that our selection of the value for the dimensionless parameter $St_{k,+}^{(el)}$ to input in the code for the simulations was perhaps not the most effective one for conveying the impact of the particle radius on that parameter. We have made modifications in the revised text (see blue color corrections on pages 3, 8 and 9) that we hope can address the concern raised by the Referee. In particular we have rewritten with more generality the characteristic conditions that the simulations must satisfy in order to lead to weak electric interactions (pages 3 and 8), and we have provided our original motivation for the selection of the values of the Stokes numbers in the simulations (page 9). As explained in our response to Referee’s comment #2, our objective in the simulations was to select values of the dimensionless

parameters to input in our code that would put the simulation results in a realistic ballpark. We also wanted our selection to enforce important physical constraints in the numerical solutions in order to isolate the effects of preferential concentration (page 9). This involved having *i*) a symmetrically dispersed case #3, for which $St_{k,+}^{(el)} = St_{k,-}^{(el)}$ and $St_{k,+}^{(ae)} = St_{k,-}^{(ae)}$ that serves as a charged baseline case, and *ii*) an asymmetrically dispersed case #2 that kept the same values of Re_λ , $St_{k,+}^{(el)}$ and $St_{k,+}^{(ae)}$ as in case #3 (i.e., same positive particle behavior in an uncoupled sense) but had different $St_{k,-}^{(ae)}$ and $St_{k,-}^{(el)}$ to induce preferential concentration while maintaining the same Stokes number ratio $St_{k,+}^{(ae)}/St_{k,+}^{(el)} = St_{k,-}^{(ae)}/St_{k,-}^{(el)}$ across the board [i.e., to keep the relative strength of hydrodynamic and electric effects on the particle clouds in Eq. (13) the same in both particle classes and both simulation cases]. This has been stressed on page 9 of the revised manuscript. Nonetheless, we must admit that the Referee is right in asserting that our choice of $St_{k,+}^{(el)}$ appears to be inconsistent with the terrestrial and Martian dust storm conditions analyzed on page 8, since $St_{k,+}^{(el)}$ in both planets is smaller than $St_{k,-}^{(el)}$ because of the inverse scaling with the radius. However, that scaling holds if one assumes that all other participating dimensional parameters pertaining to the dispersed phase are exactly the same for both classes of particles; note that this assumption is a debatable one in real dust storm conditions due to the variabilities of dimensional parameters mentioned in our response to comment #2, which is the reason why we are reluctant to commit to particular dimensional values and lose the generality of our dimensionless framework. Nevertheless, we believe (as stated on page 9 of the revised manuscript) that the overall solution of the problem does not display any significant sensitivity to $St_{k,+}^{(el)}$ once it is sufficiently small such that the electromigration velocity is much smaller than the characteristic slip velocity of the particles (see revised condition on page 3). This condition, in conjunction with $St_{k,+}^{(ae)} \gg 1$, which were already satisfied by our selection, imply that the positive charges are mostly dispersed, as explained on page 3 of the revised paper. Further decrease in $St_{k,+}^{(el)}$ to strictly meet the particle-radius scaling mentioned by the Referee would result in a slightly more dispersed distribution of positive particles, which would slightly reduce the local screening of negative charges around the negatively charged particles and would increase the characteristic length of the bulk electric field by a small amount proportional to the increase in dispersion of positive particles. We believe this change does not fundamentally modify the structure of the numerical solutions provided in our paper and therefore does not cause any significant variations in the estimate of the electric field for the Martian environment in Eq. (18). It is important for us to mention here that there is no fundamental barrier that would have impeded us to perform simulations with smaller values in $St_{k,+}^{(el)}$ in case #2 other than the computational cost associated with another simulation (i.e., $\sim 250,000$ CPU hours). Lastly, it is also worth mentioning that the $St_{k,\pm}^{(el)}$ values quoted in Eq. (16) and above have been rounded up to the second decimal digit (for the terrestrial environment) and up to the first decimal digit (for Mars), which explains the apparent discrepancy pointed out by the Referee on page 8. We would like to keep their values in that approximate form because of the reasons we have explained in our response to comment #2.

NCOMMS-17-34180 - "Aerodynamic generation of electric fields in turbulence laden with charged inertial particles" by Di Renzo & Urzay

Replies to Referee #3

We thank this Referee for the comments, which have been helpful for improving the manuscript (see revisions in **magenta color**). For convenience, we have taken the liberty of enumerating the Referee's comments.

Referee #3: 1. *This manuscript presents a scaling analysis, informed by a detailed computational investigation, on the role of particle-turbulence interactions on self-induced electric fields. Simulations of homogeneous isotropic turbulence laden with bi-disperse particles are conducted for three main cases: (i) uncharged and bidisperse, (ii) oppositely-charged and bidisperse, and (iii) oppositely-charged and monodisperse. Each particle contains a single charge that is either positive or negative. Particles of order unity Stokes number preferentially concentrate in high-strain regions of the turbulent flow, while higher Stokes number particles remain relatively uniformly distributed owing to their ballistic behavior. Because smaller particles contain an opposite charge, this segregation in particle size results in the production of long-wavelength electric fields in the carrier gas – much larger than the mean inter particle scaling or size of the smallest eddies.*

The physical scenario is very intriguing, and the analysis presents new insight into potential mechanisms responsible for atmospheric electrification. After reading this paper in detail I believe the scaling analyses and corresponding numerical approach are sound. My only concern is whether the DNS is truly representative of conditions relevant to Martian atmospheres, as this was a main focus of the study. In a rarefied atmosphere, what does Stokes ~ 1 represent? What density ratios and particle sizes would be needed to achieve this? However, the authors correctly point out the shortcomings of the approach: "Whether the range of dimensionless parameters considered above is of relevance for realistic dust storms is a question that cannot be categorically answered due to the large variabilities in flow conditions and particle properties reported in the literature, particularly for extra-terrestrial atmospheres." Moreover, I agree with their discussion later on that a point charge assumption may not be appropriate. However, I believe the authors adequately acknowledge this and the results reported here are worthy of publication. I only have a few remarks:

Authors: We appreciate the positive comments made by the Referee.

Referee #3: 2. *Despite the low number densities considered, particle accumulation due to preferential concentration in addition to Coulomb attraction may lead to a regime where particle collisions can no longer be ignored. Therefore, simply clipping the inter particle distance as is described in the methods section would miss key physics, even if the results are not sensitive to the value of the clipping distance. What are the authors' thoughts regarding this?*

Authors: The comment made by the Referee is an interesting observation that perhaps we should have addressed more thoroughly in the original manuscript. We have included additional text on page 10 (see red color text) of the revised manuscript to settle this question on quantitative grounds. To summarize the results, our observations indicate that collisions are not important in

Figure 1: Plot shown as evidence for our response to Referee’s comment #1.

our simulations. This is not to say that the regime pointed out by the Referee is not interesting (see magenta text in the last paragraph of the Discussion section to encourage future research along the same lines), but we believe the regime mentioned by the Referee would be attained at higher number densities and higher electric Stokes numbers that are outside the range needed for us to address the questions pursued in this manuscript. In the following we provide details about our observations, which have been summarized on page 10 of the revised manuscript.

As mentioned in the text, we analyzed collision frequencies in a first approximation by counting the number of times that the inter-particle distance limiter mentioned by the Referee was triggered during the simulations. The code returned a flag each time the limiter was triggered that indicated that two particles were separated at a distance smaller than $\ell_k/100$ somewhere in the flow, where ℓ_k is the Kolmogorov length. We characterized these events as necessary but not sufficient conditions for collisions. The value of the volume fraction associated with the cross section imposed by the limiter is $\phi \sim 10^{-5}$ (see also estimates of this quantity included in the examples on page 8 in the revised version). Note that this quantification exercise makes sense for particles satisfying $a_{\pm}/\ell_k \ll 1$ with a_{\pm} the particle radius. However, it should be emphasized that the limit $a_{\pm}/\ell_k \ll 1$ must necessarily be enforced in the formulation even if we were to hypothetically include collisions since the point-particle assumption requires this limit to hold in order to enable the integration of the 2nd Newton’s law of motion for the particles. In addition, the approximation $a_{\pm}/\ell_k \ll 1$ becomes increasingly more realistic as the pressure decreases since the Kolmogorov scale is larger, and as a result $2a_{\pm}/\ell_k$ rapidly attains values of order 100 in rarefied atmospheres, as in the numerical example analyzed on page 8 in the context of dust in Martian atmospheric conditions.

Figure 1 shown above provides the cumulative fraction of particles that have collided according to the criterion described above as a function time normalized with the integral time t_{ℓ} (i.e., a single

decorrelation time of the large-scale eddies). In all cases, the fraction of particles of opposite signs involved in collision events is extremely small. For instance, for case #2 in our manuscript, only 0.04% of the total number of particles time involved opposite-sign charged particles approaching each other at distances smaller than the limiter's value during one integral time. This fraction becomes even smaller according to charge transfer models, which typically state that only a small fraction of the collisions lead to charge transfer. For instance, according to Forward et al. Geophys. Res. Lett. (2009), only $(7/32) \times 0.04\% \approx 0.01\%$ of the total amount of particles would have undergone some amount of charge transfer from collisions in case #2 during one integral time. One would have to integrate this problem for more than 10,000 integral times in order to observe significant effects related to the charge transfer by collisions, which in the examples addressed in the last section of the paper (page 8 of the original manuscript) would have translated into 17 minutes of real time. This is a very long time scale that has no physical relevance for the purposes of our study. It is also worth stressing that, as shown in Figure 1 below, preferential concentration actually reduces the collision rate between oppositely charged particles by a factor of 1/4 because of the spatial segregation of particles induced by turbulence, while collisions between particles of the same sign are also suppressed due to Coloumbic repulsion, which makes collisions less relevant altogether.

Referee #3: *3. It is stated that aerodynamic forces will always dominate over Coulomb interactions, regardless of the electric Stokes number. I find this surprising, since in the limit of very large charge, I would think the second term on the right-hand side of (1) would control the particle dynamics. As likely charged particles accumulate between eddies (as shown in Fig. 1), wouldn't Coulomb repulsion act to mitigate this clustering?*

Authors: We actually agree with everything that the Referee has stated in this comment. We could not find any place in the original manuscript where we stated that aerodynamic forces will always dominate over Coulomb interactions regardless of the electric Stokes number. If we did say that in the manuscript, we would certainly disagree with our own statement and would be happy to change it since, as correctly pointed out by the Referee, the aerodynamic forces do not necessarily always dominate over Coulomb interactions. Note that the conditions under which weak electric interactions occur are provided on page 3, where we have also modified some text to make it clearer (see magenta color), and the physics corresponding to the electric mitigation phenomenon for preferential concentration mentioned by the Referee are discussed on pages 5 and 6.

Reviewer #1 (Remarks to the Author):

The authors have adequately addressed my major concern about collisions between charged grains. They now explicitly point out that triboelectric charging occurs within the saltation layer, well below the region of interest, and show that collisions between grains are indeed practically negligible in this region. After reading the revised manuscript and their detailed response, I agree that the presented results are worthy of publication.

Reviewer #2 (Remarks to the Author):

The authors have addressed the points I raised. I recommend this article for publication.

Reviewer #3 (Remarks to the Author):

The authors have adequately responded to all of my original comments. I am in favor of publication in Nature Communications.

NCOMMS-17-34180 - "Aerodynamic generation of electric fields in turbulence laden with charged inertial particles" by Di Renzo & Urzay

Final Comments

Referee #1: *The authors have adequately addressed my major concern about collisions between charged grains. They now explicitly point out that triboelectric charging occurs within the saltation layer, well below the region of interest, and show that collisions between grains are indeed practically negligible in this region. After reading the revised manuscript and their detailed response, I agree that the presented results are worthy of publication.*

Referee #2: *The authors have addressed the points I raised. I recommend this article for publication.*

Referee #3: *The authors have adequately responded to all of my original comments. I am in favor of publication in Nature Communications.*